# Stability of Leaf Yerba Mate (*Ilex paraguariensis*) Metabolite Concentrations over the Time from the Prism of Secondary Sexual Dimorphism

**DOI:** 10.3390/plants12112199

**Published:** 2023-06-02

**Authors:** Miroslava Rakocevic, Aline de Holanda Nunes Maia, Marcus Vinicius de Liz, Rafaela Imoski, Cristiane Vieira Helm, Euclides Lara Cardozo Junior, Ivar Wendling

**Affiliations:** 1Department of Research and Development, Embrapa Florestas, Colombo 83411-000, PR, Brazil; cristiane.helm@embrapa.br (C.V.H.); ivar.wendling@embrapa.br (I.W.); 2Statistical Research Group, Embrapa Meio Ambiente, Jaguariúna 13918-110, SP, Brazil; aline.maia@embrapa.br; 3Research Group on Water and Wastewater Advanced Treatment Technologies, Department of Chemistry and Biology, Federal University of Technology-Paraná, Curitiba 81280-340, PR, Brazil; marcusliz.utfpr@gmail.com (M.V.d.L.); rafaelaimoski@alunos.utfpr.edu.br (R.I.); 4Department of Pharmacy, UNIPAR, Paranaense University, Toledo 85903-170, PR, Brazil; euclideslcjr@gmail.com

**Keywords:** caffeic acid, caffeine, chlorogenic acid, phenolics, protein, theobromine

## Abstract

The yerba mate leaf metabolic composition depends mainly on genetics, sex, plant and leaf age, light intensity, harvest time, climate, and fertilization. In yerba mate, the secondary sexual dimorphism (SSD), the leaf metabolic SSD association with the frequency of leaf harvests, and the stability of the metabolites in the two genders over the years is not known. It was hypothesized that (1) the SSD in the metabolite segregation would differ among the winter and summer growth pauses, (2) females would show lower metabolite concentrations, and (3) the metabolic concentrations would show stability over the years on the same plants, not obligatorily associated with the SSD stability expression. Variations in theobromine, caffeine, chlorogenic and caffeic acids were correlated to the increasing time since the previous harvest, especially in females. However, the frequency of the metabolic SSD were associated with the studied growth pauses, rejecting the first hypothesis. No regular gender superiority was expressed in the yerba mate leaf secondary metabolites, rejecting our second hypothesis, even though more cases of superior female metabolite accumulation were identified. The stability of the leaf protein was preserved over the four years, with no SSD cases observed. The leaf methylxanthines were time stable, while the decrease in the phenolic content occurred with tree aging, which was not associated with the SSD expression, partially proving our third hypothesis. The novelty was related to the time stability of the leaf metabolic SSD observed over the winter and summer growth pauses, and over the four consecutive years without a regular expression of the male- or female-biased concentrations in the studied metabolites. To demystify the random metabolic gender responses in yerba mate, gender-orientated experiments with a high number of tree repetitions must be conducted, including clonal plants grown in various environments, such as monoculture and agroforestry, or on plantations in different climates and altitudes.

## 1. Introduction

Purine alkaloids, among which caffeine, theophylline and theobromine are the most known, represent important ingredients of beverages, including coffee, cocoa, tea, and yerba mate [1]. In plants, they act as chemical defenses against invertebrate herbivores [2]. Caffeine, theophylline, and theobromine are secondary plant metabolites from the group of purine methylxanthines that are build-up in the vacuoles of various plant species. Caffeine is mainly produced in young leaves and immature fruits, continuing to accumulate gradually during the maturation of these organs [3].

Polyphenols are grouped into two categories: flavonoids (anthocyanins, flavanols, flavanones, flavonols, flavonones, and isoflavones) and non-flavonoids (phenolic acids, xanthones, stilbenes, lignans, and tannins) [4]. A major class of phenolic plant compounds are hydroxycinnamic acids [5]. The most representative hydroxycinnamic acid is a caffeic acid, which occurs in foods primarily as an ester with quinic acid, called chlorogenic acid (5-caffeoylquinic acid, CGA). In leaves, chlorogenic acid, phenolics, and flavonoids protect the plant from oxidative stresses [6], such as excessive UV radiation exposure for the leaves in monoculture [7], diseases [8], or herbivores [9].

Leaves represent the world’s largest source of primary metabolite proteins [10] ranging from 16–29% of dry matter [11,12]. Around 80% of these proteins are concentrated in the chloroplasts where they can be divided almost equally between the soluble phase (stroma) and the system of lamellar membranes, known as thylakoids [13]. The protein extraction from green biomass generally focuses on the soluble protein fraction, which is primarily represented by the enzyme ribulose1,5-bisphosphate carboxylase/oxygenase—RuBisCO in leaves [14]. This enzyme accounts for a quarter of the nitrogen and up to half of the soluble protein in the C3 plant leaves [15].

The question is whether the previously mentioned secondary and primary metabolites could be affected by gender in dioecious yerba mate (*Ilex paraguariensis* St. Hil.), meaning that if the secondary sexual dimorphism (SSD) can be detected in the plant metabolites. Genders in dioecious flowering plants are rarely characterized by the presence of sexual chromosomes [16,17]. Therefore, the gender determination is generally possible only after the first flowering. The SSD is expressed by the sexual differences of the traits that are not directly related to gamete production [18], such as those in morphology, physiology [19], biomass production [20], or even secondary metabolites [21]. 

The yerba mate plant form, monopodial branching, and rhythmic growth correspond to Rauh’s plant architectural model [22]. Its rhythmic growth is expressed by the occurrence of two annual growth flushes forming portions of the two annual growth units, one in the spring and another in the autumn [23,24]. The rhythmic growth of yerba mate also comprises the existence of two annual growth pauses, one in the summer (total or partial phenophase of fruit ripening) and another in the winter (vegetative phenophase). The harvests of yerba mate leaves are usually performed during the growth pauses, occurring rarely in time intervals of 12 or 24 [25,26] months, and more commonly every 18 months in Brazil [27]. 

Various plant SSD characters are associated with reproduction costs, which are generally higher in females than males due to the fact that females also produce fruits and seeds. In many dioecious species, females can compensate physiologically for additional reproductive investments [28]. The SSD responses are complex and usually expressed in only certain phenophases. For example, the leaves in the female plants of yerba mate exhibit higher photosynthesis than the male plants, but only during the winter growth pause [29,30], while the two genders do not differ in the branch growth rates during any phenophase [24]. Additionally, the sexual responses of this species strongly depend on the environment, especially the light conditions, with some opposite responses observed in the female and male plants in monoculture and agroforestry [31,32].

The yerba mate leaf metabolic composition depends on genetics, sex, the plant and leaf age, light intensity, harvest time, climate, and fertilization [33,34,35,36,37,38]. More than 200 metabolic compounds are found in this species [39], with the leaves presenting higher concentrations of caffeine than theobromine and theophylline [34,39,40,41]. The SSD is expressed even in the chemical composition of the yerba mate leaves from native populations [34,42], from Brazilian clonal plants [36,43], or between clones of the two genders from Argentina [34]. Additionally, the sensory quality of the yerba mate beverage is related to the leaf chemical composition, processing methods, SSD, and even the leaf–gas exchanges [43,44,45]. The caffeine abundance is higher in male versus female plants [34], while theobromine is notably absent only in the male plants under shade, but not in monoculture [36], with a milder drink produced from monoculture composed of male plants versus female plants [44].

Some associations between the SSD with the leaf chemical composition in yerba mate are known. However, a gap exists in the leaf metabolic SSD association with the frequency of leaf harvests over phenophases, or the stability of the metabolites in the two genders over the years. The hypotheses of our work were that yerba mate (1) the SSD metabolites would vary among the harvests performed during the winter and summer growth pauses, following the ratiocination of the sexual segregation expressed in leaf photosynthesis, (2) the females would show lower methylxanthine, chlorogenic, and caffeic acid contents due to compensations for their greater reproductive costs than the males, and (3) the metabolites would show stability over the years in the same plants, which would not be obligatorily associated with the stability of the expression of SSD. To test those hypotheses, two independent experiments were executed to evaluate the SSD segregation in some of the primary and secondary metabolism components.

## 2. Results

### 2.1. Experiment 1—SSD in Some Leaf Secondary Metabolites of Yerba Mate Observed in Four Provenances during the Winter and Summer Growth Pauses

The cluster analysis and PCA of the general metabolic profiles showed the impact of the current phenophases during the three leaf harvests (Figure 1A,B). The highest normalized content of the studied components was found during the summer growth pause (G1), followed by the second growth pause (G2), while the lowest normalized content in methylxanthines, chlorogenic, and caffeic acids was found during the first growth pause –(G5; Figure 1A), with a clear segregation between the leaf winter collection after 12 months and the two posterior harvests (summer and winter, 18 and 24 months after the previous harvest; Figure 1B). The clusters corresponding to the Ivaí provenance (G3, G4, and G7) were segregated from the remaining provenances, which was coherent with the lowest studied metabolite contents of Ivaí among the four studied provenances (Figure 1A). No evidence of sexual segregation was observed in the cluster (Figure 1A) or PCA (Figure 1B) results.

The individual responses of the four studied metabolic components had some specific responses (Table 1). The theobromine and caffeine contents were the lowest in the WinP-12 phenophases in all the provenances, and there was no difference between the SumP-18 and WinP-24, with exception of Ivaí that showed some specific responses (Table 1, Figure 1A). The chlorogenic acid content was the highest in the SumP-18, followed by the WinP-24, while the lowest content was identified in the early WinP-12. Caffeic acid was the most abundant in the early WinP-12, and there was no difference between the SumP-18 and WinP-24.

In the analyses considering progeny as a factor, some cases of sexual segregation were observed (Figure 2, Appendix A). Among the four secondary metabolic components, the highest number of cases of SSD were observed in caffeine, 16 (Figure 2B); and the lowest number of cases were observed in theobromine, 10 (Figure 2A), while the SSD cases in chlorogenic and caffeic acid showed an intermediary frequency, 13 and 12, respectively (Figure 2C,D). Among them, the males were superior to the females in four, six, six, and seven cases compared to six, 12, seven, and five cases when the females were superior to the males in the theobromine (Figure 2A), caffeine (Figure 2B), chlorogenic acid (Figure 2C), and caffeic acid contents (Figure 2D), respectively.

Only the progeny 68 (from BC) repeated the SSD in theobromine content in the SumP-18 and WinP-24 (Figure 2A), but no progeny repeated the caffeine SSD over the two phenophases (Figure 2B). The progeny 1 (from Iv) and 163 (from Ca) repeated the chlorogenic acid SSD over the two phenophases (Figure 2C), while the progeny 61 repeated the caffeic acid SSD in the WinP-12 and the SumP-18 (Figure 2D). Some progenies, such as 61 (from BC) and 88 (from QI), were segregated sexually in two or three of the studied components, (5 and 21 from Iv, 53 and 68 from BC, 81 and 100 from QI, 155 and 164 from Ca).

Associated with the three growth pauses, the SSD in the theobromine content was expressed in 5:3:2 cases, the SSD in caffeine was expressed in 7:4:5 cases, the SSD in chlorogenic acid was expressed in 6:4:3 cases, and the SSD in caffeic acid was expressed in 6:6:0 cases, in the WinP-12, SumP-18, and WinP-24, respectively (Figure 2). The highest number of cases of SSD were registered in the WinP-12, and the lowest number in the WinP-24 (Figure 2 and Appendix A). However, the SSD frequency associated with the winter or summer phenophases of the leaf harvest was not significant for any studied metabolite (Table 2A). Similarly, the association of the male superiority in the theobromine, caffeine, chlorogenic, and caffeic acids was not associated with the harvest phenophases (Table 2B).

### 2.2. Experiment 2—SSD in Some Leaf Metabolites of Yerba Mate Observed in Four Provenances over Four Subsequent Years

The SSD was expressed in the theobromine and phenolic contents, but no cases of sexual metabolic segregation was observed in the caffeine or protein contents (Figure 3). The females were characterized by a higher theobromine content than the males in the QI provenance in 2015 and in the BC provenance in 2017 (Figure 3A). On the other hand, the females had a lower phenolic content than the males in the QI provenance in 2017 and a higher content in the Ivaí provenance in 2018 (Figure 3C).

The stability of the theobromine content was observed in all the provenances when the analysis with no sexual differentiation was performed, while the instability of the responses was observed in the leaves of the females from the QI provenance (Table 3). The caffeine content was not stable across the four years in the BC provenance, but when using the gender differentiation, the stability was shown. The highest instability and variation among the years was registered in the phenolic content in all the provenances, with exception of Ca. The phenolic content instability was due to both genders in Ivaí, to the males in BC, and to the females in QI. The leaf protein content was very stable in all the provenances and both genders.

### 2.3. Metabolite Correlations with Geographical Elements of the Yerba Mate Origin, Leaf Harvest Phenophase, and Plant Age

In Exp. 1, the theobromine content in the females was correlated to all the geographical/environmental elements of the yerba mate origin (i.e., altitude, latitude, and longitude), and to the increasing time of phenophases counting from the last harvest, as was also established for caffeine with the exception of latitude (Figure 4A). Methylxanthines and chlorogenic acid increased in the females as the time passed from the last harvest, while the caffeic acid content decreased. Variations in the metabolites were less correlated to the geographical provenance origin in the males than in the females, correlating only theobromine to the latitude and longitude of the provenance origin (negative), and caffeine to the longitude of the provenance origin (positive) in the former gender (Figure 4B). Caffeine and chlorogenic acid increased in the males as the time passed from the last harvest, while the caffeic acid content decreased, similar to the females. In the females, chlorogenic acid was positively correlated to methylxanthines, while the caffeic acid content was negatively correlated to all the other studied metabolites (Figure 4A). Similar responses were obtained for the males, differing only in two cases; theobromine was negatively corelated to caffeine, and chlorogenic acid was not significantly correlated to theobromine (Figure 4B).

In Exp. 2, variations in the altitude, latitude, and longitude of the provenance origin were not correlated to any of the studied metabolites (Figure 4C), while only the phenolic content diminished over time in both genders (Figure 4C,D). In this experiment, metabolite variations in the males were more correlated to the provenance geographical origin than in the females, positively correlating caffeine to the altitude of the provenance origin and negatively to the longitude in the former (Figure 4D). The theobromine content was negatively correlated to the caffeine content, while protein was correlated to caffeine in the females (Figure 4C). No cases of significant correlation among the studied metabolites was observed in the males (Figure 4D).

## 3. Discussion

The novelty of our research was related to the time stability of the leaf metabolic SSD of yerba mate observed over the winter and summer pauses of growth, and over four consecutive years without a regular expression of male- or female-biased concentrations of the studied metabolites. 

The variations in the theobromine, caffeine, chlorogenic, and caffeic acids were correlated to increasing time that passed from the previous harvest, especially in the females (Figure 4A). However, the frequency of the metabolic SSD was not associated with the studied phenophases of the leaf harvest (Table 1), rejecting the first hypothesis. Similarly, the SSD in the metabolic profiles of *Baccharis dracunculifolia* did not manifest, while the leaf age significantly impacted the metabolic profiling [46]. The yerba mate leaves were emitted in phenophases that corresponded to the growth flushes preceding the growth pauses studied here, and they remained alive on branches rarely longer than 7–9 months in monoculture, and at a maximum of 19–21 months in agroforestry [34]. The growth rhythmicity could essentially explain the difference between the summer harvest (leaves emitted during the spring flush) *versus* the winter harvest (leaves emitted during the autumn flush), which were very clearly expressed in the chlorogenic acid content (Table 1). On the other hand, the caffeine and theobromine contents were the lowest during the winter pause after 12 months since the previous harvest when the highest caffeic acid content was observed. The significant negative correlation between the caffeine and theobromine contents was found in the males of Exp. 1 (Figure 4B) and the females of Exp. 2 (Figure 4C), which could be related to the synthesis of the two methylxanthines. Caffeine is synthesized through multiple methylations of xanthine (initial purine compound) derivatives, with theobromine as the intermediator in the biosynthesis pathway [47], which could explain the negative correlation between caffeine and theobromine.

When the analyses of Exp. 1 were conducted using the phenophases and provenances, the highest mean contents of the studied metabolites were observed in the summer (Figure 1, Table 1). Those responses had a similar trend to those previously observed [25]. Meanwhile, no SSD was observed using multivariate analyses (Figure 1). When the progenies were considered in the analyses, the metabolic SSD was expressed, but without the regular appearance of the same progenies that were sexually different in the three observed phenophases of any of the four studied metabolites (Figure 2, Appendix A). The yerba mate caffeine concentrations ranged from ~0.5 to 2.5%, and theobromine between 0.3 and 0.7%, both varyingly dependent on the plantation origin, indicating a sexual differentiation in the cluster analyses that were dependent on genotype [48]. Considering the absolute concentrations, the leaf contents of caffeine and theobromine were in previously indicated ranges (Figure 2A,B and Figure 3A,B). Similarly, theobromine was not detected in the male plants cultivated under the shade of agroforestry. However, its content was higher in the former under clearings during both the spring and winter harvests [36]. The responses obtained in our experiment conducted in a monoculture showed theobromine differences in phenophases (Figure 1, Table 1), with no regular prevalence for any gender or phenophase (Figure 2, Table 2). Only a higher female than male theobromine content was identified in the provenance QI (2015) and BC (2017) (Figure 3A). The obtained responses compared to those of Pauli et al. [36] indicated genotype and climate modifications in the theobromine gender responses.

In the needles of the female plants of *Juniperus communis*, a higher concentration of carbohydrates, carbon, and phenolic compounds were found than in their male counterparts, indicating that the females allocated more resources to storage and defense than the males [49]. In wild *Dioscorea* spp., female plants are superior in their accumulation of various secondary metabolites and possess a superior antimicrobial and antioxidant potential compared to their male plant counterparts [50]. No regular gender superiority was expressed in the yerba mate leaf secondary metabolites (Table 2), rejecting our second hypothesis, even though more cases of a superior female metabolite accumulation were identified (Figure 2).

The male reproductive structures of *Pseudotsuga menziesii* required more nitrogen for pollen production, while the females required more carbon to finish the reproductive cycle during the seed development [51]. The females of *Juniperus thurifera* grew more and stored more non-soluble sugars than the males, especially under low-stress conditions [52]. On the other hand, a high leaf nitrogen concentration was usually positively correlated with the photosynthetic rates, related to an increased enzyme concentration involving photosynthesis, especially RuBisCO [53]. The yerba mate leaf proteins from the J-protein/HSP40 family (determined about 140) assisted with plant survival during multiple stresses for plants cultivated under a monoculture, while also promoting the plant’s growth under forest shade [54]. Any depletion of foliar nitrogen due to the reproductive nitrogen demand of organs developed in the leaf proximity could reduce foliar photosynthesis, confounding the increased photosynthesis associated with a reproductive carbon sink [49]. In yerba mate plants, the total leaf nitrogen and carbon were greater in the female clones than in the male counterparts at the end of flowering [32]. In our experiment, the leaf protein content did not show any sexual differentiation in any year or provenance (Figure 3D and Table 3), which was potentially related to its adaptation to the environment and genotype modulations of the protein gender responses.

The phenolic, caffeic, and chlorogenic acid contents strongly differed between the genders in *Rumex thyrsiflorus*, always with the female-biased concentrations, leading to the idea that the female plants of this species could be more stress-tolerant compared to the males [55]. In our experiment, the SSD was also observed in the phenolic, caffeic, and chlorogenic acid contents (Figure 2C,D and Figure 3C), but the regular male- or female-biased amount was not detected (Table 2). As the SSD is species dependent, the contents of phenolics, caffeic, and chlorogenic acid did not differ between the two genders in *Baccharis trimera* and *B. myriocephala*, while a strong impact of the geographic origin was observed [56]. The geographic origin, here the longitude of the provenance origin that could be interpreted as the distance from the ocean and a more continental climate (Figure 5), was correlated to yerba mate theobromine and caffeine. Meanwhile, the concentrations of all the studied metabolites increased as the time passed from the last harvest, with an exception for the caffeic acid that diminished in Exp. 1 (Figure 4A,B and Table 1). In Exp. 2, only the phenolic accumulation in the leaves diminished with the plant age in both genders (Figure 4C,D, Table 3), and showed the SSD without the detection of any constant gender superiority (Figure 3C). Among all the studied metabolites, the stability over the four consecutive years was the greatest in the protein content, and the most unstable in the phenolic compounds, with the last related to both the male and female variations over the four subsequent years of harvest (Table 3). The phenolic compounds in yerba mate showed stability when the leaf ages of one and six months were compared [35], while we observed an instability over various tree ages harvested every year, but not related to the leaf age. The previous results partially proved our third hypothesis that the gender response stability was preserved in some metabolites over the years and was not regularly associated with the instability of the expression of SSD over the years, with the last specially related to the phenolic compounds.

The SSD was not expressed in the studied primary metabolites of yerba mate leaves (proteins) but was detected in all the studied secondary metabolites (theobromine, caffeine, chlorogenic, caffeic, and phenolic compounds). The studied secondary metabolites provided the defense roles in plants, and some dioecious species were associated with a higher female accumulation, such as in *Rumex thyrsiflorus* [55] or *Populus cathayana* [57]. Male-biased cases were also detected, such as in *Juniperus oxycedrus macrocarpa* [58]. The gender association to such secondary leaf metabolite accumulations can provide increased gender resistance to environmental pressure [55] and resistance to herbivores [57], which seems to be random in yerba mate, where random metabolic gender responses appeared to be prevalently related to the local tree environmental pressure and the tree age only in the phenolic compounds (Figure 4C,D). To demystify the random metabolic gender responses in yerba mate, special gender-orientated experiments with a high number of tree repetitions must be conducted, including clonal plants grown in various environments, such as monoculture and agroforestry [32] or on plantations in different climates and altitudes. 

## 4. Materials and Methods

### 4.1. Experimental Field, Provenances, and Sequence of Harvests in Experiments 1 and 2

For the evaluation of the yerba mate half-sibling progenies (seedlings originating from the same mother tree), an experiment was installed in a monoculture (full sunlight cultivation), including various progenies originated from seven provenances. The experiment was implemented in Ivaí (25°01′39″ S, 50°51′32″ W, 748 m.a.s.l.), Paraná state, Brazil, in November 1997. The planting arrangement of the seedlings, all produced in Ivaí, was a row distance of 3 m and a plant-to-plant distance in the row of 2 m. The Ivaí climate type, according to Köppen-Geiger’s classification, was Cfa, corresponding to a humid subtropical climate with hot summers, infrequent frosts, and the tendency of rains concentrated in the summer months without a defined dry season [59]. The means of the minimum, average, and maximum annual temperatures were 12.7, 17.8, and 24.9 °C, respectively. The mean annual rainfall wats 1588 mm and he mean annual relative humidity was 79.7%. The soil was rhodic hapludox, comprising 72% clay, and was acidic with a low base saturation, a high aluminum saturation [60], and the relief was smoothly wavy with slopes around 4% [61].

The studied yerba mate progenies originated from the following four provenances: Ivaí (Iv) 25°01′39″ S, 50°51′32″ W, 748 m.a.s.l., Barão de Cotegipe (BC) 27°37′15″ S, 52°22′48″ W, 765 m.a.s.l., Quedas do Iguaçú (QI) 25°27′ S 52°54′28″ W, 630 m.a.s.l., and Cascavel (Ca) 24°57′20″ S, 53°27′19″ W, 782 m.a.s.l. (Figure 5). The first harvest was caried out in the winter of 1999, followed by two subsequent harvest held every 24 months (2001 and 2003). 

Two independent experiments of the yerba mate metabolomic composition were conducted. The first experiment (Exp. 1) was focused on three phenophases that followed the harvest of 2003. They corresponded to the (1) winter growth pause that happened 12 months after the previous pruning on August 10, 2004 (WinP-12); (2) the summer growth pause occurring 18 months after pruning on November 30, 2004 (SumP-18); and (3) the winter growth pause occurring 24 months after pruning on June 14, 2005 (WinP-24). The harvests were performed on different blocks and trees, but always on the same provenance/progenies. For the second experiment (Exp. 2) that focused on the stability of the gender metabolic contents, mature leaves with no injuries were collected in August every year for four years during the winter growth pause from 2015–2018, when the plants were 18–21 years old, respectively. The harvests were performed always on the same trees at a frequency of 12 months.

In both experiments, the mature leaves were collected from the medium third parts of the trees. The content of two methylxanthines, the ones mostly presented in yerba mate leaves (theobromine and caffeine), were determined alongside chlorogenic and caffeic acid analyses in Exp. 1 and total phenolics and total protein analyses in Exp. 2.

In Exp. 1, the following progenies were used. The progenies identified as 1, 3, 4, 5, 7, 8, 10, 11, 15, 21, and 25 originated from provenance Iv, the progenies 51, 53, 57, 58, 59, 61, 65, 68, 69, and 70 originated from BC, the progenies 80, 81, 84, 86, 87, 88, 91, 92, 93, and 100 originated from QI, and the progenies 151, 152, 155, 157, 159, 162, 163, 164, 165, and 171 originated from Ca. The number of female (F) and male (M) trees (replicates) of each progeny in each of three phenophases was between one and five. 

In Exp. 2, the following progenies were used. The progenies identified as 8, 11, and 25 originated from provenance Iv, the progenies 51, 52, 56, 58, 59, 61, 64, and 65 from BC, the progenies 76, 81, 86, 92, 93, 96, and 100 from QI, and the progenies 151, 152, 155, 157, 159, 162, 163, 164, 165, and 171 from Ca. The number of female trees from each provenance ranged from three to six and the male trees from two to seven. The trees within each gender class were considered replicates in each of the four harvest years.

### 4.2. Metabolite Determination

In Exp. 1, approx. 100 g of leaves from each plant were boiled in water for 10 s, and the green and clean leaves were dried in a forced draft oven (45 °C, 48 h). Afterwards, the leaves were ground, placed on paper bags, wrapped with plastic, and refrigerated until the chemical analyses. The caffeine, theobromine, chlorogenic, and caffeic acid contents were first extracted from the leaves (1 g) by maceration in 50 mL of methanol: water (70:30, *v*/*v*), and were then filtered using 0.45 μm nylon filters. Finally, chromatographic analyses were performed using high-performance liquid chromatography (HPLC, Shimadzu Mod. SCL-10A, Kyoto, Japan). A 5 μm Supelcosil LC C18 analytical column with the dimensions of 4.6 × 250 mm was used (Sigma-Aldrich Inc., USA). The detection was monitored at 265 nm for caffeine and theobromine, and at 325 nm for caffeic and for methylxanthines and chlorogenic acids using an SPD-10A UV–vis detector (Shimadzu, Kyoto, Japan) coupled to the chromatograph. For further details about the extraction and detection procedures, see Cardozo-Junior et al. [25].

In Exp. 2, the leaf samples were dried in a microwave oven (power 1500 W, frequency 2450 MHz) for 4 min, alternating the position of leaves at 60 s intervals for homogenous drying. Subsequently, the leaves were crushed, sieved at 0.5 mm, packed, and stored in a freezer (−20 °C). Lately, the aqueous extracts that contained 0.1 g of plant material were prepared in 50 mL of ultrapure water type I heated to its boiling temperature (100 °C under a pressure of 1 atm) to determine the contents of caffeine, theobromine, and the total phenolic compounds. Then, the samples were homogenized using an ultrasound (Ultracleaner 1400 A, Unique, Indaiatuba, Brazil) for 30 min, cooled to room temperature, filtered with a complete volume of 100 mL in a volumetric flask, and frozen (−20 °C). 

To determine the methylxanthine (caffeine and theobromine) contents in Exp. 2, the extracts were thawed and manually homogenized for 30 s. Approx. 2 mL of the extract was filtered in a 0.22 μm membrane with a syringe and a holder. Next, an aliquot was transferred to a 1.5 mL amber vial with a Teflon cap. These samples were injected directly into the HPLC (Agilent 1260 Infinity model, Agilent^®^, Santa Clara, CA, USA) controlled by ChemStation software (Agilent^®^, Santa Clara, CA, USA) and equipped with a G1311B quaternary pump, a G1329B automatic injector, and a DAD G4212B detector (Agilent^®^, Santa Clara, CA, USA). The Acclaim 120 C18 column (2.1 × 150 mm, Dionex^®^, Thermo Fisher Scientific, Waltham, MA, USA) and an Acclaim C18 (2.1 mm, 5 μm guard cartridge, Thermo Fisher Scientific, Waltham, MA, USA) were used to separate the compounds. The conditions used to separate the compounds from the aqueous extract (10 μL injection) were 30 °C with a flow of 0.3 mL min^−1^ of eluent with the mobile phase A (H_2_O: acetic acid—99.5:0.5 *v/v*) and B (Merck^®^ acetonitrile—100%). The wavelength used to detect the compound was 280 nm (fixed). The gradient elution program was 0–8 min (4% B), 8–12 min (4–5% B), and 12–30 min (5% B). 

The quantification of the total phenolic compounds in Exp. 2 was conducted according to the Folin–Ciocalteau spectrophotometric method [62] with modifications. A measure of 0.1 mL of the extract, 6.0 mL of distilled water, and 0.5 mL of the Folin–Ciocalteau reagent were added to a volumetric flask and stirred for 1 minute. Afterwards, 2 mL of a 15% Na_2_CO_3_ solution was added and stirred for another 30 s. The final volume was adjusted with distilled water to 10 mL. The reaction was kept in the dark at room temperature for two hours and, subsequently, the absorbances were recorded in a spectrophotometer at 760 nm. An analytical curve was obtained with the total phenolic compounds in gallic acid (3,4,5-trihydroxybenzoic acid) between the concentrations of 0.25 and 13 mg L^−1^ (R^2^ = 0.9988), and the results expressed in mg were equivalent to the total phenolic compounds per gram of the dry sample (mg GAE g^−1^). 

The determination of the total protein content in Exp. 2 followed the micro-Kjeldahl method, using the total nitrogen content multiplied by the universal conversion factor 6.25 [63].

### 4.3. Experimental Design and Statistical Analyses

Completely randomized designs (CRD) were adopted for both Exp. 1 and Exp. 2. In all the experiments, the experimental unit was a single plant, from which leaves were collected for the determination of the four metabolic compounds. In both experiments, the data were unbalanced regarding the number of female and male plants within progeny or within provenance due to the gender determination in the adult tree years.

The dendrogram construction used the means for the leaf metabolic compounds determined in Exp. 1 (theobromine, caffeine, chlorogenic acid, and caffeic acid) that were calculated by phenophase (WinP-12, SumP-18, and WinP-24), provenance (Iv, BC, QI, and Ca), and gender (F, M), resulting in a 4 × 24 matrix. It was based on a cluster analysis, allowing for the visualization of the similarities among 24 categories of the four metabolites. The similarity analysis allowed for the identification of the main factors (phenophase, provenance, or gender) that were responsible for the separation of the categories. A principal component analysis (PCA) also helped to identify those factors. Both multivariate methods were applied to the standardized data to avoid an unbalance in the contribution of each variable to the group formation due to their different magnitudes. In the clustering process, the Euclidean distance was used as the metric to determine the separation between the clusters and Ward’s minimum variance was used as the clustering method. The standardization of each individual variable for the cluster and PC analyses was performed by dividing the original values by their respective standard errors. The cluster analysis and the PCA were performed using the ‘CLUSTER’ and ‘PRINCOMP’ procedures, respectively, while the dendrogram was constructed using the ‘TREE’ procedure in SAS/STAT^®^ [64].

In both experiments (Exp. 1 and Exp. 2), various types of ANOVA were applied using an F-test for the contrasts of gender/phenophase/progeny/provenance. The ANOVA’s assumption of normality was not critical because the F-tests were robust enough to lack of normality [65], especially whenever the analyzed variables were continuous, and the model residues did not present outliers. The most important assumptions that needed to be checked were the homogeneity of the variances and the random errors not correlated, where the latter was guaranteed by the experimental design [66]. In Exp. 1, two types of two-way ANOVAs were performed. The first was applied to investigate the influence of gender on the response variables within the progeny. In those cases, the treatments were formed by combinations of the progeny factors and the plant gender (female, male), resulting in twelve completely randomized k × 2 factorial designs (k progenies varying from 18 to 22 and two genders), one for each phenophase (WinP-12, SumP-18, and WinP-24) and provenance (Iv, BC, QI, and Ca). The second was applied to investigate the influence of the phenophase within the provenance without accounting for the plant gender, resulting in a CRD 3 × 4 factorial design. The influence of gender on the leaf contents for each of the four studied metabolites, as well as the comparisons among the phenophases, were examined via F-tests for the contrasts, using the GLM (general linear model) procedure of the SAS/STAT^®^ [64]. In Exp. 2, sixteen one-way ANOVAs were performed to assess the influence of the plant gender on the response variables within each year (2015–2018) and provenance (Iv, BC, QI, and Ca). In this experiment, it was not possible to examine the gender effect within the progeny due to the limiting number of trees per progeny with metabolite data. The significance of the ‘gender’ factor within the provenance and harvest year (2015–2018) was assessed directly via the ANOVA F-test, as this factor was composed of only two levels and factorial arrangements were not adopted in this case. 

Aiming to verify whether the frequency of the SSD occurrence would be similar among the summer and winter growth pauses (1st hypothesis), a contingency table analysis was performed via Fisher´s exact test (using the FREQ procedure of the SAS/STAT^®^ [64]) for each studied metabolite. The same test was applied to investigate whether the frequency of the progenies presented a yerba mate male metabolite superiority over phenophases (second hypothesis).

To verify the stability of some leaf metabolites over the years (third hypothesis), linear regression models of the GLM procedure (SAS/STAT^®^ [64]) were used. The pooled data from all the trees were considered. The linear models were fitted by gender, performing the adjustment and estimation of the respective parameters. Whenever the statistical hypothesis of the null slope coefficient was not rejected, the scientific hypothesis of the stability was accepted. 

The correlations among the provenance’s geographical/ecological characterization (altitude, latitude, and longitude), time/age (phenophases by months 12, 18, and 24; age from 18–21 years old trees for 2015–2018), and studied metabolites were performed using and the ‘corrplot’ package of the statistical software R [67]. The adopted significance level was 0.10 due the high within treatment (or group) variability, as indicated by the ANOVA coefficients of variation.

## 5. Conclusions

The influence of the winter and summer growth pauses on the SSD frequency in the yerba mate leaf theobromine, caffeine, and chlorogenic acid contents was not observed, rejecting our first hypothesis about a higher metabolite SSD frequency during the winter growth pauses. Meanwhile, a higher chlorogenic acid content was observed in the summer growth pause. The metabolite SSD was expressed over phenophases and years, occurring randomly in various provenances and progenies, irregularly showing male or female metabolite content superiority, and rejecting our second hypothesis regarding a male metabolite superiority. The stability of the leaf protein was preserved over the four years, with no one SSD case observed. The leaf methylxanthine contents were stable over the four subsequent years, while a decrease in the phenolic leaf content was observed with plant aging, which was not associated with the SSD expression, partially proving our third hypothesis. The gender association with the secondary metabolite accumulations in the leaves seemed to be random in yerba mate. Those random metabolic gender responses appeared to be related to local tree environmental pressure and age (only in phenolic compounds). Special gender orientated experiments with a high number of tree repetitions grown in various environments or plantations established in different climates or altitudes will need to be conducted to demystify the apparently random metabolic gender responses identified in yerba mate.

## Figures and Tables

**Figure 1 plants-12-02199-f001:**
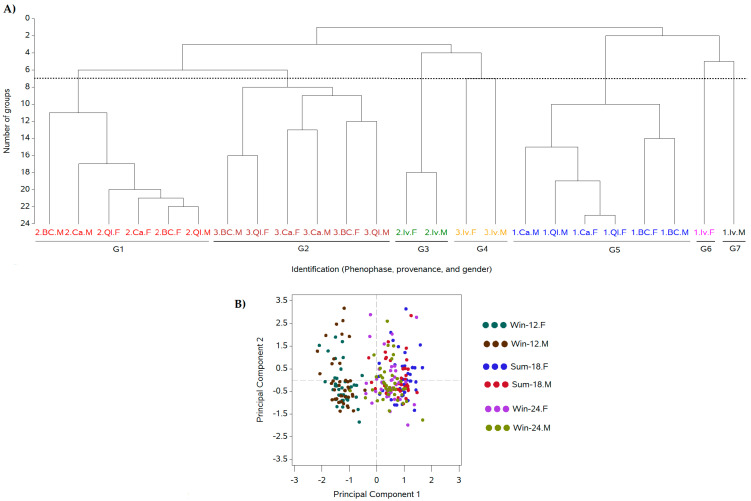
Similarity among the normalized metabolic profiles: (**A**) dendrogram and (**B**) PCA considering theobromine, caffeine, chlorogenic, and caffeic acids in the leaves of female (F) and male (M) plants from four provenances (Iv, BC, QI, and Ca), with leaf collections performed in three phenophases: (1) The winter growth pause (WinP-12), (2) The summer growth pause (SumP-18), and (3) The second winter growth pause (WinP-24), during harvests performed at 12, 18, and 24 months after pruning, respectively. The components PC1 and PC2 accounted for 46.04% and 28.96% of the overall data variability, resulting in a cumulative variance of 75.01%.

**Figure 2 plants-12-02199-f002:**
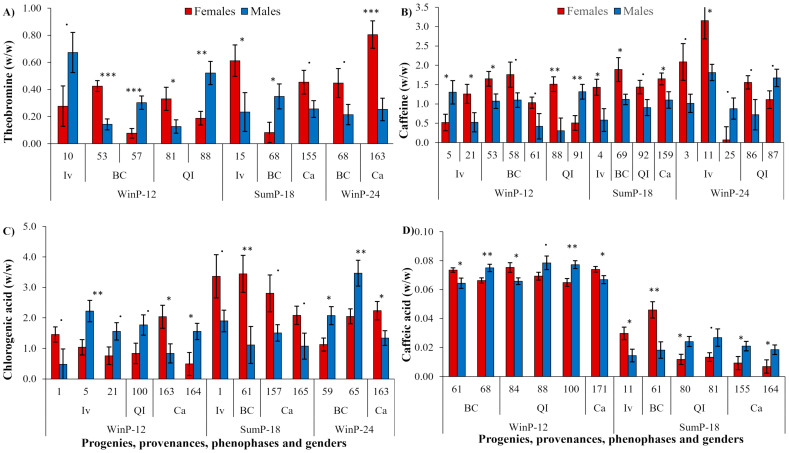
Dynamics in the leaf (**A**) theobromine, (**B**) caffeine, (**C**) chlorogenic acid, and (**D**) caffeic acid concentrations (%, *w*/*w*), considering the cases of the expressed secondary sexual dimorphism. The plants originated from four provenances in Southern Brazil: Ivaí (Iv), Barão de Cotegipe (BC), Quedas do Iguaçú (QI), and Cascavel (Ca); and the leaf collections were performed over three phenophases: the winter growth pause, WinP-12; the summer growth pause, SumP-18; and the second winter growth pause, WinP-24 that occurred at 12, 18, and 24 months after pruning, respectively. The SSD presence as quantified by the significance level of the ANOVA F test (for n = 1–5) for the gender factor (‘***’ < 0.001, ‘**’ < 0.01, ‘*’ < 0.05, ‘.’ < 0.10) were also indicated.

**Figure 3 plants-12-02199-f003:**
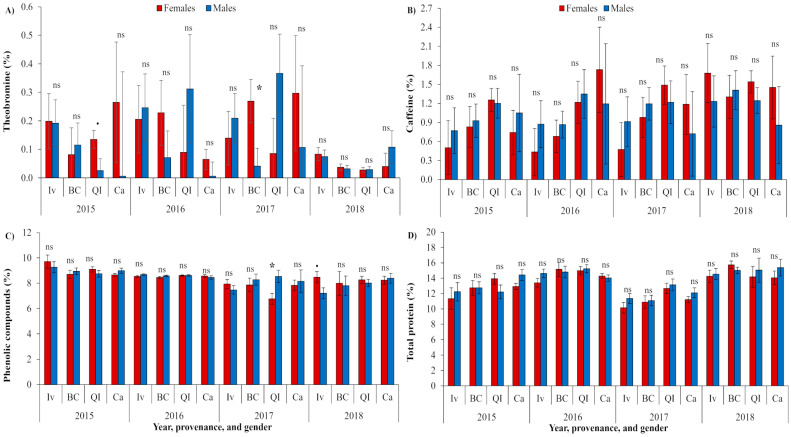
Dynamics in the leaf (**A**) theobromine, (**B**) caffeine, (**C**) phenolic compounds, and (**D**) total protein contents (%) considering that the male and female plants originated from four provenances (Iv, BC, QI, and Ca, in Southern Brazil), and were observed over four years, 2015–2018, corresponding to 18–21 years-old trees. The significance of the ANOVA F-test (n = 2–7) for the gender factor (‘*’ < 0.05, ‘^.^’ < 0.10) was also indicated.

**Figure 4 plants-12-02199-f004:**
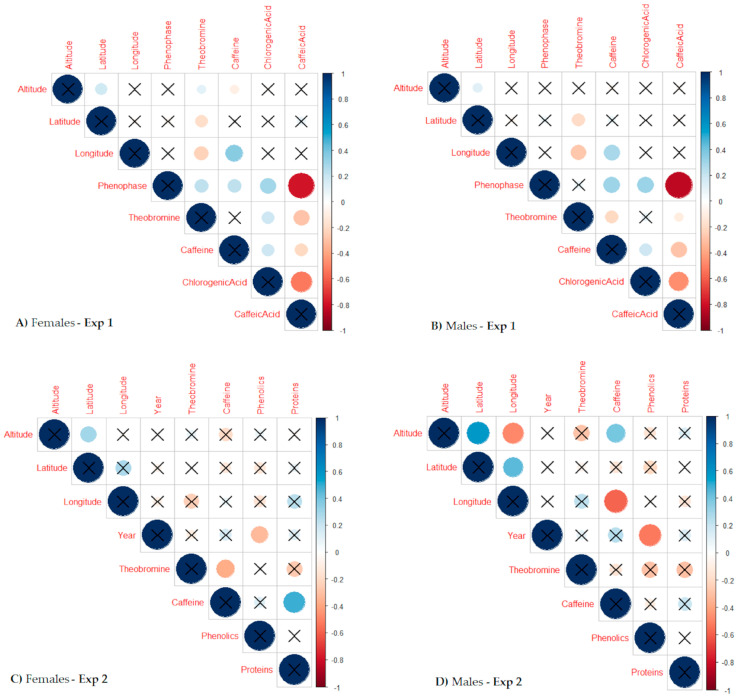
Graphical presentation of the coefficients (values corresponding to the circle size, colors, and color intensities) and *p*-values (significant when not crossed circles, n = 1–5, *p* < 0.10) for the correlations among the geographic characteristics of four provenances in Southern Brazil, considering their altitude (QI: 630 m, Iv: 748 m, BC: 765 m and Ca: 782 m), latitude (Ca: 24.91, Iv: 25.02, QI: 25.45 and BC: 27.61), and longitude (Iv: 50.86, BC: 52.38, QI: 52.91 and Ca: 53.91), with leaf harvests performed 12, 18, or 24 months after the previous harvest (**A**,**B**), or corresponding to the plant age (18, 19, 20, and 21 years-old trees (**C**,**D**)) and analyzed metabolites in the two experiments (Exp. 1 and Exp. 2) in the yerba mate females and males.

**Figure 5 plants-12-02199-f005:**
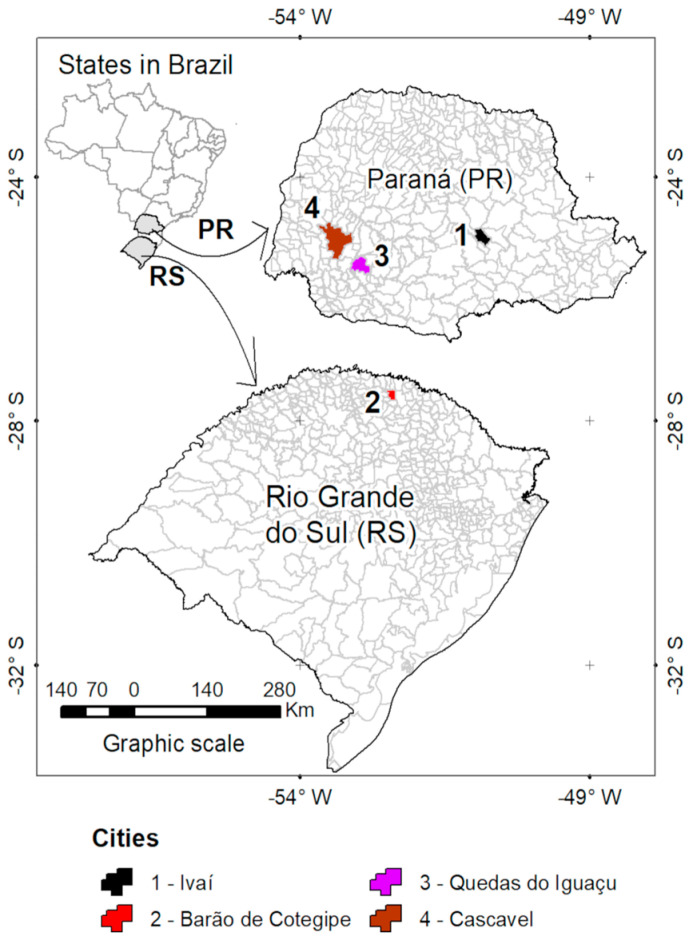
Schematic representation of the geographical characteristics (latitude, longitude, altitude, and location) of the four provenances of yerba mate progenies and the location of the respective municipalities in the states of Paraná and Rio Grande do Sul, Southern Brazil.

**Table 1 plants-12-02199-t001:** Estimated means and respective standard errors (SE), n = 52–60, for the contents of theobromine, caffeine, chlorogenic, and caffeic acids (%, *w*/*w*) in the leaves of yerba mate plants *.

		Estimated Means ± SE Corresponding to Each Provenance			
Metabolite	Phenophase	Ivaí	Barão de Cotegipe	Quedas do Iguaçú	Cascavel
Theobromine (%, *w/w*)	WinP-12	0.33 ± 0.02 ^a^	0.12 ± 0.02 ^b^	0.16 ± 0.02 ^b^	0.16 ± 0.02 ^b^
	SumP-18	0.35 ± 0.02 ^a^	0.23 ± 0.02 ^a^	0.22 ± 0.02 ^a^	0.23 ± 0.02 ^a^
	WinP-24	0.34 ± 0.02 ^a^	0.20 ± 0.02 ^a^	0.23 ± 0.02 ^a^	0.27 ± 0.02 ^a^
Caffeine (%, *w/w*)	WinP-12	0.70 ± 0.06 ^c^	1.02 ± 0.06 ^b^	1.18 ± 0.06 ^b^	1.13 ± 0.06 ^b^
	SumP-18	0.96 ± 0.06 ^b^	1.30 ± 0.06 ^a^	1.35 ± 0.06 ^a^	1.35 ± 0.06 ^a^
	WinP-24	1.11 ± 0.06 ^a^	1.31 ± 0.06 ^a^	1.35 ± 0.06 ^a^	1.45 ± 0.06 ^a^
Chlorogenic acid (%, *w/w*)	WinP-12	1.09 ± 0.08 ^c^	1.09 ± 0.08 ^c^	1.10 ± 0.08 ^c^	1.13 ± 0.08 ^c^
	SumP-18	2.44 ± 0.08 ^a^	2.45 ± 0.08 ^a^	2.60 ± 0.08 ^a^	2.40 ± 0.08 ^a^
	WinP-24	1.60 ± 0.08 ^b^	1.80 ± 0.08 ^b^	1.66 ± 0.08 ^b^	1.79 ± 0.09 ^b^
Caffeic acid (%, *w*/*w*)	WinP-12	0.068 ± 0.001 ^a^	0.067 ± 0.001 ^a^	0.068 ± 0.001 ^a^	0.068 ± 0.001 ^a^
	SumP-18	0.012 ± 0.001 ^b^	0.013 ± 0.001 ^b^	0.012 ± 0.001 ^b^	0.012 ± 0.001 ^b^
	WinP-24	0.012 ± 0.001 ^b^	0.015 ± 0.001 ^b^	0.014 ± 0.001 ^b^	0.012 ± 0.001 ^b^

Means followed by the same letters in the columns, corresponding to each metabolite, were not significantly different. * Plants were originated from four provenances, namely, Ivaí (Iv), Barão de Cotegipe (BC), Quedas do Iguaçú (QI), and Cascavel (Ca), with leaf harvests performed during three phenophases, 12, 18, and 24 months after pruning, corresponding to the first winter growth pause (WinP-12), the summer growth pause (SumP-18), and the second winter growth pause (WinP-24), respectively. The number of replicates (number of trees per phenofase) from each provenance were Iv (55–60), Bc (55–57), QI (53–54) and Ca (52–58), depending on the phenophase. The MSE (mean square error) degrees of freedom of the two-way ANOVA was 656.

**Table 2 plants-12-02199-t002:** Relative frequencies (%) of yerba mate progenies from four provenances in in Southern Brazil, regarding each leaf metabolite and expressing (A) the secondary sexual dimorphism (SSD) * or (B) the mean male over the mean female superiority **.

Factor	Metabolite	Factor Class	% of Progeny with ‘YES’ Cases	Fisher’s Exact Test *p*-Value *
(A) SSD expression	Theobromine	Winter pause	9.68	0.7513
		Summer pause	6.82	
	Caffeine	Winter pause	18.09	0.4547
		Summer pause	11.36	
	Chlorogenic acid	Winter pause	13.83	0.7913
		Summer pause	11.36	
	Caffeic acid	Winter pause	7.45	0.3475
		Summer pause	13.64	
(B) Mean male over mean female superiority	Theobromine	WinP-12	60.00	0.7143
		SumP-18	33.33	
		WinP-24	00.00	
	Caffeine	WinP-12	28.57	0.6154
		SumP-18	0.00	
		WinP-24	40.00	
	Chlorogenic acid	WinP-12	66.67	
		SumP-18	0.00	0.1072
		WinP-24	66.67	
		WinP-12	50.00	
	Caffeic acid	SumP-18	66.67	1.0000

* Comparison between the winter and summer growth pauses. ** Comparison among the winter growth pause, WinP-12; the summer growth pause, SumP-18; and the second winter growth pause, WinP-24, occurring 12, 18, and 24 months after pruning, respectively.

**Table 3 plants-12-02199-t003:** Estimates with the respective standard errors (±SE) of slopes (β) of the provenance-specific linear regression models, used as the metrics to investigate the stability of the yerba mate leaf metabolites in trees from four provenances in Southern Brazil: Ivaí (Iv), Barão de Cotegipe (BC), Quedas do Iguaçú (QI), and Cascavel (Ca).

Metabolite	Provenance	Overall Model ^(a)^		Gender-Specific Model ^(b)^		
		Parameter Estimate	*p*-Value ^(c)^	Gender	Parameter Estimate	*p*-Value
Theobromine	Iv	−0.04 ± 0.03	0.1459	F	−0.04 ± 0.03	0.2138
				M	−0.04 ± 0.04	0.3991
	BC	−0.02 ± 0.03	0.4987	F	−0.01 ± 0.05	0.8537
				M	−0.03 ± 0.03	0.2899
	QI	−0.01 ± 0.04	0.7485	F	−0.03 ± 0.01	**0.0165**
				M	0.01 ± 0.09	0.8967
	Ca	−0.02 ± 0.06	0.6904	F	−0.04 ± 0.08	0.6018
				M	0.04 ± 0.04	0.3680
Caffeine	Iv	0.19 ± 0.13	0.1499	F	0.28 ± 0.21	0.1850
				M	0.13 ± 0.18	0.4706
	BC	0.17 ± 0.08	**0.0493**	F	0.17 ± 0.15	0.2594
				M	0.17 ± 0.10	0.1104
	QI	0.06 ± 0.08	0.4465	F	0.11 ± 0.09	0.2556
				M	0.00 ± 0.15	0.9943
	Ca	0.07 ± 0.16	0.6618	F	0.17 ± 0.21	0.4202
				M	−0.10 ± 0.19	0.7319
Phenolics	Iv	−0.58 ± 0.12	**<0.0001**	F	−0.41 ± 0.19	**0.0450**
				M	−0.75 ± 0.15	**<0.0001**
	BC	−0.33 ± 0.15	**0.0308**	F	−0.27 ± 0.21	0.2126
				M	−0.37 ± 0.21	**0.0904**
	QI	−0.40 ± 0.14	**0.0084**	F	−0.55 ± 0.20	**0.0142**
				M	−0.23 ± 0.18	0.2422
	Ca	−0.20 ± 0.12	0.1155	F	−0.22 ± 0.17	0.2180
				M	−0.16 ± 0.09	0.1426
Proteins	Iv	0.45 ± 0.32	0.1623	F	0.55 ± 0.46	0.2423
				M	0.39 ± 0.45	0.3932
	BC	0.37 ± 0.33	0.2598	F	0.47 ± 0.56	0.4195
				M	0.31 ± 0.41	0.4615
	QI	0.13 ± 0.31	0.6877	F	−0.17 ± 0.34	0.6307
				M	0.55 ± 0.58	0.3591
	Ca	0.06 ± 0.35	0.8694	F	−0.07 ± 0.44	0.8775
				M	0.23 ± 0.56	0.6986

^(a)^ Linear regression models fit using data from both female and male trees; ^(b)^ linear regression models fit separately for each tree gender; ^(c)^ nominal significance level associated with the hypothesis of stability (β = 0). * Leaf harvests were conducted over four subsequent years (2015–2018), corresponding to 18–21 years-old trees in the four provenances (Iv, BC, QI, and Ca). The values marked in bold were significant (*p* < 0.10), indicating a lack of stability. The number of trees used to estimate the regression parameters ranged from seven to 13 per harvest year.

## Data Availability

All the original data and performed calculations can be obtained from the authors.

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
