# Peer review of "Stability of Leaf Yerba Mate (*Ilex paraguariensis*) Metabolite Concentrations over the Time from the Prism of Secondary Sexual Dimorphism"

_plants, 2023, doi:10.3390/plants12112199_

Round 1

Reviewer 1 Report

1. What is the main question addressed by the research?  The objective of this research was to know the effect on the metabolite content in relation to the association of secondary sexual dimorphism in yerba mate, the harvest frequency (winter and summer) and the stability of these metabolites in two genera of this plant.
2. Do you consider the topic original or relevant in the field? 
Yes
Does it address a specific gap in the field? 
Yes
3. What does it add to the subject area compared with other published material? 
Novelty was related to time stability of leaf metabolic SSD observed over the winter and summer growth pauses, and over the four consecutive years, without regular expression of male- or female-biased concentrations of studied metabolites.
4. What specific improvements should the authors consider regarding the methodology? What further controls should be considered? 
A special gender orientated experiment with high number of tree repetitions grown in various environments or plantations established in different climates, or altitudes, must be conducted to demystify the apparently random metabolic gender responses in yerba-mate.
5. Are the conclusions consistent with the evidence and arguments presented and do they address the main question posed? 
The conclusions are consistent with the arguments presented and address the main question posed, the rejected hypotheses of the work are indicated.
6. Are the references appropriate? 
Minor changes are required, information is described in the Comments and Suggestions for Authors section
7. Please include any additional comments on the tables and figures.

 Information requested is described in the Comments and Suggestions for Authors section.

The manuscript is new and interesting, however, it is necessary to follow up on the following comments:

Line 52: delete space… [11,12].

Line 63: delete space… [16,17].

Line 70: delete space… [23,24].

Line 74: delete space… [25,26].

Line 80: delete space… [29,30].

Line 84: delete space… [31,32].

Line 86: modify format… [33–38].

Line 88: modify format… [34,39–41].

Line 89: delete space… [34,42].

Line 90: delete space… [36,43].

Line 92: modify format… [43–45].

Line 119: Did you mean Figure 1A instead of Figure 1a?

Line 129: it is usually recommended that figures and tables appear in the text as close possible to the first paragraph in which they are mentioned.

Line 144: Table titles should be brief, more detailed information should appear at the bottom of the table.

Line 150: the letters indicating significant differences must appear in superscript text format.

Line 163: in some bars of figure 2, deviations higher than the average value appear, is this correct?

Line 184: Table titles should be brief, more detailed information should appear at the bottom of the table.

Line 209: in some bars of figure 3, deviations higher than the average value appear, is this correct?

Line 239: Table titles… Idem

Line 280: avoid using author name information, follow indication through the document

Line 370: insert space… 24.9 °C

Line 407: correct… 45 °C…. 48 h

Line 445: correct Na2CO3

Line 446: 30 s

Line 546: use 407–413 instead of 407-13

Line 548: use 407–413 instead of 407-13

Line 593: use 217–225 instead of 217-225

Line 603: use 149–151 instead of 149-151

Line 609: use 1453–1461 instead of 1453-1461

Line 615: use 4835–4842 instead of 4835-4842

Line 623: use 71–83 instead of 71-83

Line 627: use 569–579 instead of 569-579

Line 632: use 1100–1103 instead of 1100-1103

Line 658: correct… 29(2), 135

Line 663: use 279–286 instead of 279-286

Author Response

Dear Reviewer,

Thank you for your suggestions. We resected all, and they are marked in the text with a red color.

The manuscript is new and interesting, however, it is necessary to follow up on the following comments:

Line 52: delete space… [11,12].

            Done

Line 63: delete space… [16,17].

Done

Line 70: delete space… [23,24].

Done

Line 74: delete space… [25,26].

Done

Line 80: delete space… [29,30].

Done

Line 84: delete space… [31,32].

Done

Line 86: modify format… [33–38].

Done

Line 88: modify format… [34,39–41].

Done

Line 89: delete space… [34,42].

Done

Line 90: delete space… [36,43].

Done

Line 92: modify format… [43–45].

Done

Line 119: Did you mean Figure 1A instead of Figure 1a?

Yes, thank you. Corrected.

Line 129: it is usually recommended that figures and tables appear in the text as close possible to the first paragraph in which they are mentioned.

The problem here is the size of the text and Figures. If made by the general rule, the Figure 1 would stay alone at one page, followed by only one or two paragraphs alone at the following page. If the publisher would like to change this order, they are free to make their format.

Line 144: Table titles should be brief, more detailed information should appear at the bottom of the table.

Done.

Line 150: the letters indicating significant differences must appear in superscript text format.

Done.

Line 163: in some bars of figure 2, deviations higher than the average value appear, is this correct?

Yes, some cases had very high deviation.

Line 184: Table titles should be brief, more detailed information should appear at the bottom of the table.

Done.

Line 209: in some bars of figure 3, deviations higher than the average value appear, is this correct?

Yes, some cases had very high deviation.

Line 239: Table titles… Idem

Done.

Line 280: avoid using author name information, follow indication through the document

Corrected.

Line 370: insert space… 24.9 °C

Corrected.

Line 407: correct… 45 °C…. 48 h

Corrected.

Line 445: correct Na2CO3

Corrected.

Line 446: 30 s

Corrected.

Line 546: use 407–413 instead of 407-13

Corrected.

Line 548: use 407–413 instead of 407-13

Corrected.

Line 593: use 217–225 instead of 217-225

Corrected.

Line 603: use 149–151 instead of 149-151

Corrected.

Line 609: use 1453–1461 instead of 1453-1461

Corrected.

Line 615: use 4835–4842 instead of 4835-4842

Corrected.

Line 623: use 71–83 instead of 71-83

Corrected.

Line 627: use 569–579 instead of 569-579

Corrected.

Line 632: use 1100–1103 instead of 1100-1103

Corrected.

Line 658: correct… 29(2), 135

Corrected.

Line 663: use 279–286 instead of 279-286

Corrected.

Reviewer 2 Report

Authors presented studies considering the effect of sexual dimorphism of Ilex paraguariensis (yerba mate) on the concentration of metabolites in leaves. The time of harvest and the stability of the tested compounds were also taken into account. This subject of research seems to be a scientific novelty as well as important from the consumer’s point of view. Please clarify a few points in the methodology before publishing.

1. Please unify the significant numbers.

2. Please correct the numbering of sections.

3. Has the normality of the data distribution been studied?

4. Figure 1B. Could you explain the scale of the PCA plot? The PCA plots are usually presented in the range -1.0 to 1.0.

5. Figure 1B. What criteria were used to assess the number of significant components? There are two components (PC1, PC2) on the plot without describing the variability of the results. Please add the variability (%) which is explained by these components.

6. Tables 1 and 3. How many replicates (or samples) were used to calculate the mean and standard error?

7. Lines 432-434. Agilent is an American company (Santa Clara, CA).

8. Line 506-508. Please clarify how the high variability between tree was determined. How was the significance level (p=0.1) determined?

Author Response

Authors presented studies considering the effect of sexual dimorphism of Ilex paraguariensis (yerba mate) on the concentration of metabolites in leaves. The time of harvest and the stability of the tested compounds were also taken into account. This subject of research seems to be a scientific novelty as well as important from the consumer’s point of view. Please clarify a few points in the methodology before publishing.

Thank you for your suggestions and questioning. Our responses are marked in blue, over the manuscript, to facilitate the localization of responses/explanations.

  1. Please unify the significant numbers.

Done

  1. Please correct the numbering of sections.

Done

  1. Has the normality of the data distribution been studied?

The ANOVA´s assumption of normality is not critical because the F tests are robust to lack of normality, especially whenever the variables analyzed are continuous and the model residues do not present outliers. The most important assumptions to be checked are homogeneity of variances and random errors not correlated (this last one is guaranteed by the experimental design).

In the manuscript, the explanation was done in lines 487-493.

  1. Figure 1B. Could you explain the scale of the PCA plot? The PCA plots are usually presented in the range -1.0 to 1.0.

The principal components PC1 and PC2 are linear components of the standardized (normalized) original variables. The standardization of each individual variable is performed by dividing the original values by their respective standard errors. Consequently, the ranges of the standardized values for each variable are not limited to -1.0 to 1.0 (depends on the original data variability). Therefore, linear combinations of these variables are not limited to the (-1,1) interval.

In the manuscript, the short explanation was done in lines 482-483.

  1. Figure 1B. What criteria were used to assess the number of significant components? There are two components (PC1, PC2) on the plot without describing the variability of the results. Please add the variability (%) which is explained by these components.

The percentage of the overall variability explained by the components PC1and PC2 were 46.04% and 28.96%, respectively, resulting in a cumulative variability of 75.01%. We used only two components to allow the 2D visualization, considering the expressive amount of variance captured by PC1 and PC2 together (75.01%).

In the manuscript, the declaration was done in lines 142-144.

  1. Tables 1 and 3. How many replicates (or samples) were used to calculate the mean and standard error?

The number of replicates used in Exp1 (Table 1) is not the same for all combinations of provenance versus phenophase – it ranged from 52 to 60, totaling 668 trees. In Table 3, The number of trees used to estimate the regression parameters ranged from 7 to 13 per harvest year. This required information was included in the footnotes of both Tables 1 and 3. We also added the degrees of freedom of the ANOVA´s mean square errors, which is the most important information related to the power of the statistical tests.

In the manuscript, the details were done in lines 153-155 and 240-250.

  1. Lines 432-434. Agilent is an American company (Santa Clara, CA).

Corrected

  1. Line 506-508. Please clarify how the high variability between tree was determined. How was the significance level (p=0.1) determined?

We modified this phrase “The adopted significance level was 0.10, due the high between tree variability…” To “The adopted significance level was 0.10, due the high within treatment (or group) variability as indicated by the ANOVA´s coefficients of variation” (lines 523-525).
